# Diet-induced steatohepatitis does not cause heart failure with preserved ejection fraction in male middle-aged C57BL/6N mice

**Chase W. Kessinger**[☉], **Lin Chen**[iD][☉], **Felicia Huang, Jierui Lin, Amanda Davenport, Zhiqiang Lin**[iD][☉]*

Department of Biomedical Research and Translational Medicine, Masonic Medical Research Institute, Utica, New York, United States of America

☉ These authors contributed equally to this work.
* zlin@mmri.edu

## Abstract

Metabolic dysfunction-associated steatohepatitis (MASH) is a condition characterized by inflammation and liver fibrosis. MASH patients are at a high risk of developing heart failure with preserved ejection fraction (HFpEF), thus raising the question of whether liver dysfunction directly or indirectly leads to HFpEF. Diet-induced murine liver disease models are well established; however, it is not clear whether these mice develop HFpEF. In this work, we studied the metabolic pathophysiological effects of two formulated diets, including a high fat (HF) diet, and a high fat, high fructose, high cholesterol diet (HFFC). Both the HF and the HFFC diets induced obesity and liver steatosis, with the HF diet being more potent in increasing adipose tissue volume and the HFFC diet being more potent in increasing hepatocyte lipid storage. Additionally, the HFFC diet, not the HF diet, caused liver inflammation and fibrosis. Although the HFFC diet induced steatohepatitis in treated mice, these mice had normal cardiac function and histology. In conclusion, our data demonstrate that HFFC diet-stress induces MASH without harming the heart and suggest that liver dysfunction is not sufficient to cause HFpEF in mice.

## Introduction

Metabolic dysfunction-associated fatty liver disease (MAFLD), previously known as non-alcoholic fatty liver disease (NAFLD) [1], is a chronic liver disease with its prevalence increasing worldwide [2]. MAFLD is associated with profound systemic disturbances in lipid metabolism [3], and patients with MAFLD are at risk of progressing to steatohepatitis (MASH), a condition characterized by inflammation and fibrosis in the liver. MAFLD patients are more likely to have cardiovascular comorbidities, such as hypertension, stroke, coronary artery disease and heart failure [4,5].

**Data availability statement:** All relevant data are within the manuscript.

**Funding:** Z. L. was supported by National Heart, Lung, and Blood Institute (NHLBI) (HL146810), Taconic research grant and American Heart Association (AHA) Transformational Project Award (25TPA1478779). C.W.K was supported by NHLBI (HL158816) and MMRI institutional fund. The funders had no role in study design, data collection and analysis, decision to publish, or preparation of the manuscript.

**Competing interests:** The authors have declared that no competing interests exist.

Heart failure is a condition characterized by the inability of the ventricles to fill or eject blood, resulting in inadequate cardiac output. It can arise from changes in the structure or function of the myocardium, valves, and vessels of the heart, and is clinically classified as two distinct forms: heart failure with reduced ejection fraction (HFrEF) and heart failure with preserved ejection fraction (HFpEF) [6]. Although many medicines for managing HFrEF are available, only a few therapeutics treating HFpEF have been developed in recent years (see review [7]). Among the many risk factors, MASH is closely associated with HFpEF [8]. In a retrospective study analyzing the echocardiography data of biopsy-validated MAFLD population, MASH was found to be associated with impaired myocardial relaxation and enlarged left atrium [9], both of which are HFpEF diagnostic parameters [6].

Animal models are critical for defining the molecular mechanisms that link MASH and HFpEF. In mice, HF diet stress induces liver steatosis, but HF diet alone does not induce HFpEF. Therefore, a two-hit HFpEF mouse model has been developed, which uses L-NAME (a drug that induces hypertension) and a HF diet to treat adult male mice for up to 15 weeks [10]. This mouse model recapitulates most of the clinical features of HFpEF and has been widely used since its publication. Recently, another murine HFpEF model has been developed via expressing renin in the liver of metabolic dysfunction mice [11]. In these two HFpEF models, the introduction of chronic hypertension is essential for the metabolic dysfunctional mice to develop HFpEF.

The relationship between HFpEF and MASH has been studied in a high fat/high cholesterol diet-induced hamster MASH model, which shows a clear correlation between steatohepatitis and HFpEF [12]; however, the lack of genetically modified hamster strains limits the use of this hamster MASH model for studying the molecular mechanisms linking MASH with HFpEF. In contrast, a large collection of gene modified mouse lines are publicly available and are an invaluable asset for deciphering the genetic factors controlling disease progression, thus it is a high impact question to determine whether murine MASH model could be used for studying HFpEF. Besides genetic influence, sex is another risk factor for MASH. For example, middle-aged male patients more likely to develop MASH than age-matched female patients [13], and this sex dimorphism is recapitulated in diet-induced murine MASH models [14,15]. For this scientific reason, we chose to use male mice in the current study.

To establish a vigorous diet-induced MASH model for studying HFpEF, we compared the pathological effects of two formulated diets for triggering MASH, including a high fat (HF) diet, and a high fat, high fructose, high cholesterol diet (HFFC, also known as AMLN diet [16]). We found that both the HF diet and the HFFC diet treatment induced liver steatosis, but only the HFFC diet caused MASH. By characterizing the cardiac phenotypes of the HF diet- and the HFFC diet-treated mice, we concluded that neither liver steatosis nor MASH caused HFpEF in middle-aged male C57BL/6N mice.

## Materials and methods

### Mice

All animal procedures were approved by the Institutional Animal Care and Use Committee (IACUC) of Masonic Medical Research Institute. Protocol approval number: 2024-09-27-ZL02. C57BL/6N mice were provided by Taconic Biosciences. Mice were sacrificed with $CO_2$ overdose.

### Diet stress and tissue collection

HF diet containing 60 kcal% fat (D12492) and HFFC diet (D09100310) containing 40 kcal% fat (Mostly Palm Oil), 20 kcal% fructose and 2% cholesterol were purchased from Research diets Inc. The diet stress was started with 8-weeks-old male C57BL/6N mice, and they were sacrificed for tissue collection 36 weeks after diet stress. For each cohort, 8 animals were included in the beginning of the study. For tissue collection, mice were first sacrificed with $CO_2$ overdose, and cardiac perfusion was immediately followed to remove blood from the tissues. For the following studies, 4–6 successfully collected liver or heart samples were included in the histology and molecular analysis.

### Gene expression

Tissues were lysed in TRI reagent (Zymo Research). RNA was extracted with Direct-zol RNA Miniprep Kits (Zymo Research). For quantitative reverse transcription PCR (qRT-PCR), RNA was reverse transcribed with all-in-one 5X RT Master Mix (Applied Biological Material, Cat# G512) and transcripts were measured using SYBR Green and normalized to *Gapdh*. Primers used in this study are listed in S1 Table.

### Western blot

Protein was extracted from the TRI reagent following the direction of the Direct-zol RNA Miniprep Kits. Briefly, the flow-through after the RNA binding to the column was precipitated with 4 volumes of cold acetone (−20°C). After centrifuge, the pellet was washed with 95−100% ethanol and then dissolved in RIPA buffer supplemented with 1% SDS. Protein concentration was measured by Pierce BCA Protein Assay Kit (Cat# 23225), and 20 µg total protein was loaded into SDS PAGE gel for western blot. Primary antibodies used in this study: MYH6 antibody (ABclonal A9516), GAPDH antibody (Proteintech, 60004–1-lg).

### Histology

For F-actin and BODIPY staining, livers were fixed in 4% PFA, cryoprotected with 30% sucrose and embedded in OCT. 10 µm thickness cryosections were incubated with Phalloidin-iFluor™ 555 (Cayman 20552, 1:200) and BODIPY 493/503 (Cayman 20552, 1:200) for 2 hours at room temperature. After three times washing with PBS, stained sections were mounted in water and immediately used for imaging with a Zeiss LSM-700 confocal microscope. For hematoxylin and eosin (H&E) and picrosirius red staining, the tissues were fixed with 4% PFA, dehydrated and embedded in paraffin. After staining, imaging was performed on a Keyence BZ-X800 microscopy system.

### Cardiac ultrasound

Cardiac ultrasound was performed with a Vevo 3100 preclinical imaging system. For conscious echocardiography, mice were first trained for three days before being imaged. Briefly, mice were held by standard handhold. The chest was cleared of hair with a hair remover. The transducer was placed on the chest to perform short-axis mid papillary muscle M-mode measurements of the heart. The animal was held for about 1 minute at a time, then put down after proper echocardiography images were acquired. This was repeated two or three times per mouse. For mitral valve Doppler echocardiography, mice were anesthetized with isoflurane (1–5%) and placed in dorsal recumbency on the heated ultrasound

imaging platform. During the imaging process, the platform and transducer were manipulated to produce the proper imaging angles and planes for a complete cardiac ultrasound analysis. After imaging, the mice were placed back in the holding cage and returned to the colony once ambulatory.

### Micro-CT imaging

Micro-CT whole-body imaging was performed on a Quantum GX, with a field of view of 72 mm, a voxel size of 144 μm, a voltage of 50 kV, and a current of 160 μA. Visceral adipose tissue (VAT) and Subcutaneous adipose tissue (SAT) volumes were segmented and quantified from whole-body 3D datasets using Analyze 15 software.

### Statistics

Normally distributed data values (e.g., body weight, gene expression) were expressed as mean ± SD and analyzed with Student's t-test (two groups) or one-way ANOVA followed by Tukey's post hoc test (more than two groups). Non-parametric data (e.g., cell size, lipid droplet size) were analyzed using the Mann-Whitney test (two groups) or Kruskal-Wallis test followed by Dunn's multiple comparisons (more than two groups). Prism 10 software was used to plot bar/violin graphs and to perform statistical analysis.

## Results

### The HF diet is more effective than the HFFC diet to induce obesity

Both the HF diet and the HFFC diet have been used to induce metabolic stress in mice; however, the pathological effects of these diets have not been directly compared. To address this question, we treated 8-week-old male C57BL/6N mice with either of these two diets for 36 weeks (Fig 1A). Age-matched Chow diet-fed male C57BL/6N mice were used as controls. Although both the HF and the HFFC diets increased body weight (Fig 1B), the HF diet was more effective than the HFFC diet to cause obesity (Fig 1C), which was not due to diet preference as evidenced by the similar HF and HFFC diet uptake rate (Fig 1D). Tibia bone length was not distinguishable between the Chow, HF, and HFFC diet fed mice (Fig 1E), suggesting that developmental variances of these three cohorts do not cause the body weight differences. Both the HF diet and the HHFC diet induced hyperglycemia (Fig 1F), suggestive of metabolic dysfunction in these mice.

### Subcutaneous adipose tissue expansion is a signature of the HF-diet induced obesity

Our data suggest that the HF diet is more effective than the HFFC diet in inducing obesity, a condition characterized by adipose tissue expansion. To further corroborate this observation, we used Micro-computed tomography (Micro-CT) technology to measure adipose tissue volumes (Fig 2A). Micro-CT datasets were analyzed to separate subcutaneous adipose tissue (SAT) from visceral adipose tissue (VAT) based on Hounsfield units and physical location (Fig 2B). Compared to mice fed a Chow diet ($9.22 \pm 2.26$ cm$^3$), the adipose tissue volumes of mice fed with either the HF diet or the HFFC diet were increased, with the HF diet-fed mice containing significantly more adipose tissue than the HFFC diet-fed mice ($25.53 \pm 1.34$ cm$^3$ vs $15.88 \pm 0.94$ cm$^3$, Fig 2C). By quantifying the SAT depots, we found that HF diet tripled the SAT volume (HF $18.02 \pm 2.97$ vs Chow $5.71 \pm 1.55$ cm$^3$), whereas the SAT volume of the HFFC diet-fed mice was mildly increased ($10.17 \pm 1.87$ cm$^3$) (Fig 2D). Similar with the SAT, the VAT volume of the HF diet-fed mice was the highest ($7.51 \pm 0.89$ cm$^3$), followed by the HFFC ($5.71 \pm 0.93$ cm$^3$) VAT and Chow diet VAT ($3.51 \pm 0.73$ cm$^3$) (Fig 2E). Among these three cohorts mice, the HF diet-fed mice had a significantly higher SAT to VAT ratio, and this ratio was not distinguishable between the Chow and the HFFC diet-fed mice (Fig 2F).

Collectively, these data demonstrate that the HF diet is more effective than the HFFC diet in inducing adipose tissue expansion.

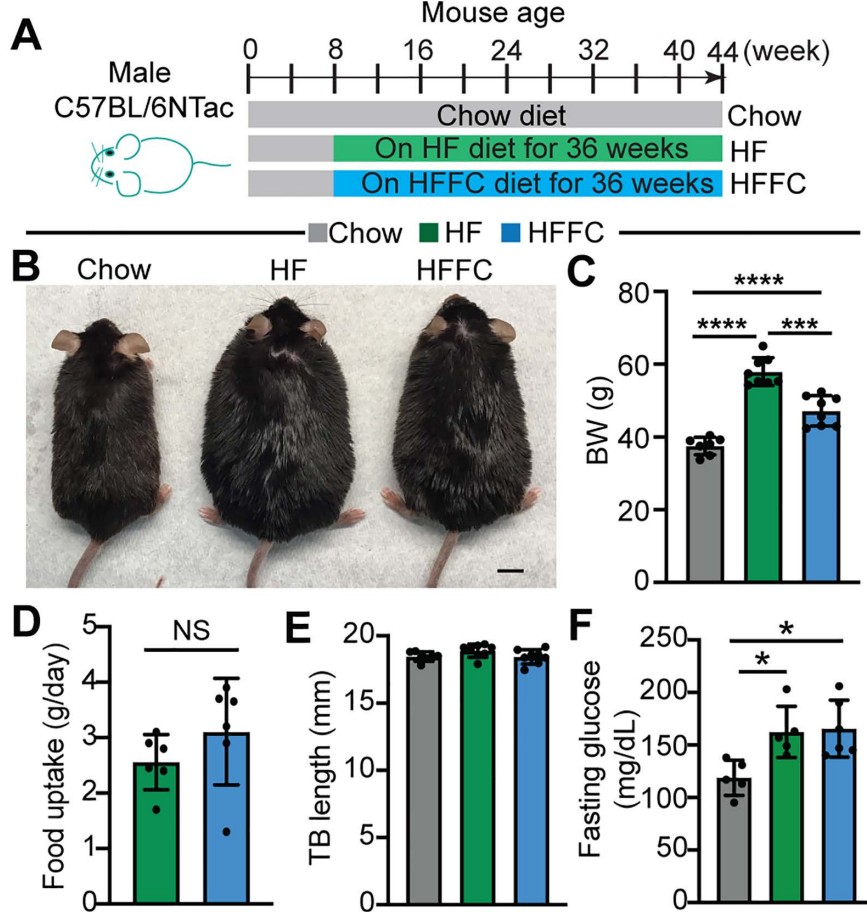

**Fig 1. Chronic high fat diet treatment induces obesity. A**. Experimental design. Age-matched male mice were treated with different diets. **B**. Appearance of the mice 36 weeks after the HF diet or the HFFC diet treatment. Scale: 1 cm. **C**. Body weight (BW) measurements. **D**. Tibia bone (TB) length. C and D, each group, n = 8. **E**. Food uptake per mouse per day. Each group, n = 6. **F**. Baseline fasting serum glucose level. N = 5. C and F, one-way ANOVA test, *, p < 0.05.

### The HFFC diet is more potent than the HF diet to induce hepatic steatosis

Compared to the Chow diet, both the HF and the HFFC diets increased liver size, with the HFFC diet showing the most substantial hepatomegaly effect (Fig 3A and 3B). This observation raised the question of whether the HF and the HFFC diets increased liver size by enlarging hepatocyte size. Additionally, the HFFC liver showed a lighter color than the other two livers (Fig 3A), suggesting that the HFFC liver might have more lipids deposited in the hepatocytes. To decipher possible cellular changes, we then examined hepatocyte size and cellular lipid droplet size. Fluorophore-conjugated phalloidin and BODIPY were used to stain the filamentous actin cytoskeleton, enriched in the cellular cortex, and lipid droplets, respectively [17].

Phalloidin staining clearly revealed the hepatocytes borders (Fig 3C), enabling us to measure their surface areas. In line with the liver size data (Fig 3A), both the HF and the HFFC hepatocytes were enlarged, and the HFFC hepatocytes were significantly larger than the HF hepatocytes (Fig 3C and 3D). To determine whether the cell size enlargement was caused by lipid deposition, we measured the sizes of BODIPY-positive lipid droplets. In the control hepatocytes, the lipid droplets were too small to be measured; in the HF and the HFFC hepatocytes, large lipid droplets were formed and

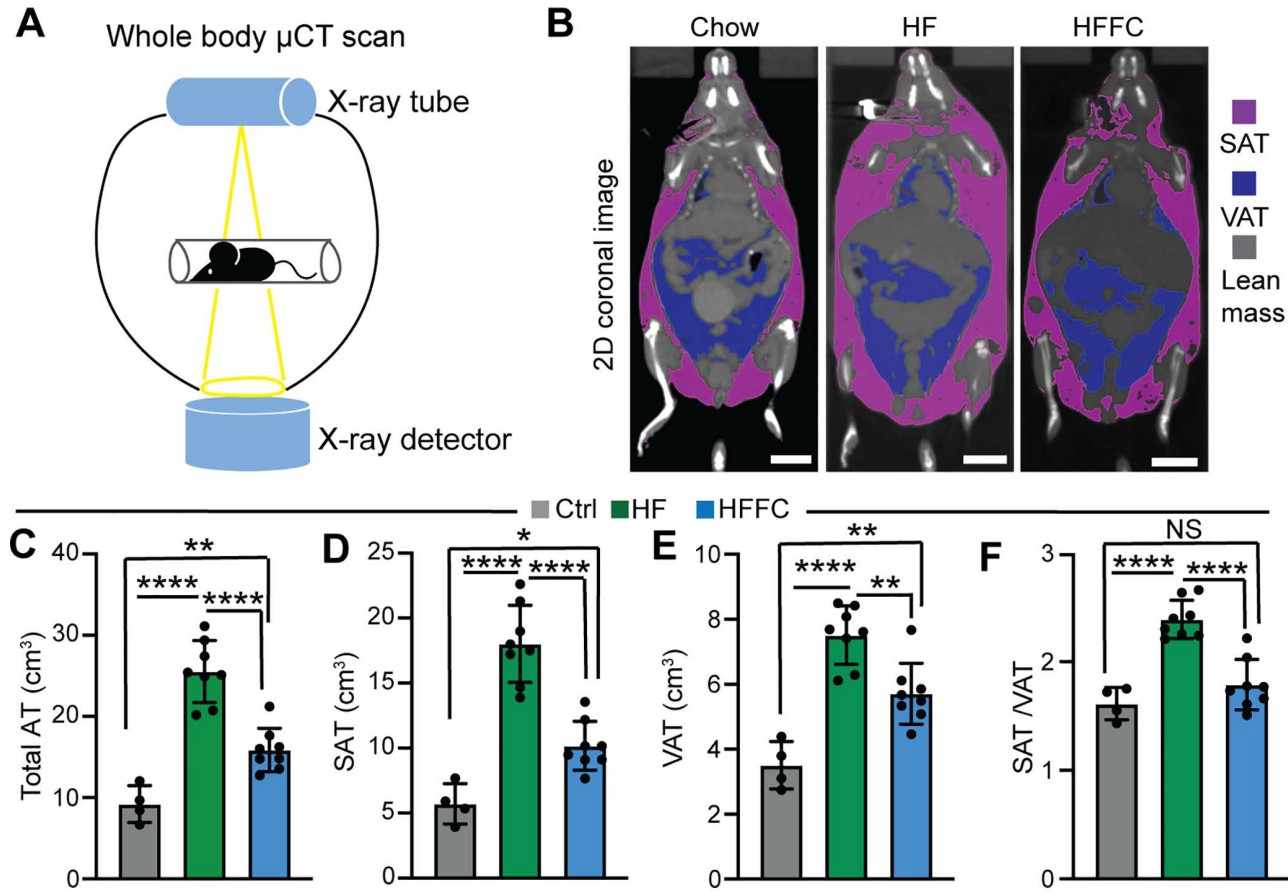

**Fig 2. Micro-CT analysis of adipose tissue volume. A**. A schematic view of the Micro-CT scan setting. X-ray projections were acquired over a 360° rotation around the mice. The individual X-ray projections were then reconstructed to produce 2D cross-sectional images and 3D volumes. **B**. Representative 2D coronal images showing the adipose tissue composition. Scale: 10 mm. **C**. Total adipose tissue volume. **D**. Subcutaneous adipose tissue (SAT) volume. **E**. Visceral adipose tissue (VAT) volume. **F**. SAT to VAT ratio. C, D, E, and F, Chow diet group, n = 4; HF diet group, n = 8; HFFC diet group, n = 8. One-way ANOVA test, *, p < 0.05; **, p < 0.01, ****, p < 0.0001. NS, not significant.

therefore were measurable (Fig 3C). By quantifying the hepatocyte lipid droplets surface areas, we found that the lipid droplets in the HFFC hepatocytes were significantly larger than the HF hepatocytes lipid droplets (Fig 3E).

Together, these data suggest that the HF- and the HFFC-diet- induced liver size enlargement is due to lipid deposition-mediated hypertrophic growth of hepatocytes, and that the HFFC diet has stronger hepatic steatosis effects than the HF diet.

### The HFFC diet and not the HF diet causes MASH

We further tested whether the HF diet and the HFFC diet damaged the liver and affected heart function in the following studies. Thirty-six weeks of HFFC diet treatment caused extensive hepatic steatosis, hepatocyte ballooning (Fig 4A), and pericellular fibrosis (Fig 4B); however, the same period of HF diet stress only caused hepatic steatosis (Fig 4A and 4B). On the molecular level, the HFFC diet robustly increased the expression of *ColA1* (Fig 4C), a marker gene for fibrosis [18], and activated the expression of pro-inflammatory cytokine genes, such as *Il-1b* [19], *Icam1* [20], and *Ccl2* [21] (Fig 4D and 4E). In contrast to the HFFC diet, the HF diet did not significantly increase the expression of *ColA1* or the three

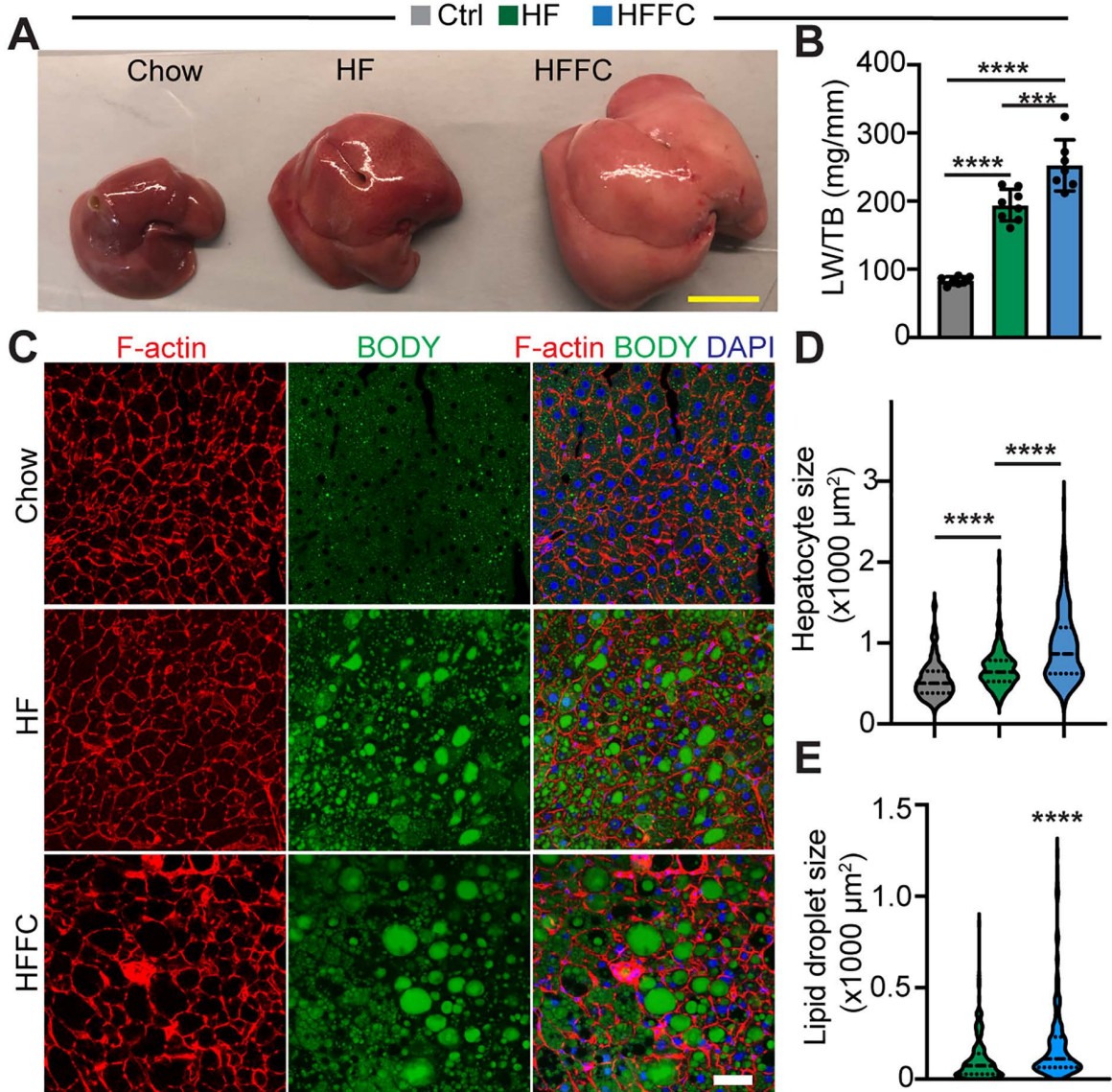

**Fig 3. Both the HF diet and the HFFC diet stress causes liver steatosis.** **A**. Liver morphology. Scale: 1 cm. **B**. Liver weight (LW) to Tibia bone (TB) length ratio. One-way ANOVA test, ***, p < 0.001; ****, p < 0.0001. Each group, n = 8. **C**. Representative fluorescence images of liver sections. Scale: 50 µm. **D**. Hepatocyte size measurement. For each group, 200 hepatocytes from 4 animals were measured. Kruskal-Wallis test, ****, p < 0.0001. **E**. Lipid droplet size measurement. 200 Lipid droplets from the hepatocytes of four animals were measured. Mann-Whitney test, ****, p < 0.0001.

pro-inflammatory cytokine genes (Fig 4D and 4E). These data together confirmed that the HFFC diet rather than the HF diet treatment caused MASH.

### Neither the HF diet nor the HFFC diet treatment alters cardiac diastolic function

In hamsters, HFFC diet treatment induced MASH and caused HFpEF [12]. We then asked whether HFFC-induced MASH mice also developed HFpEF. First, we performed M-mode echocardiography in conscious mice to assess cardiac systolic function at the end of the study. HFFC diet treatment did not change cardiac fraction shortening

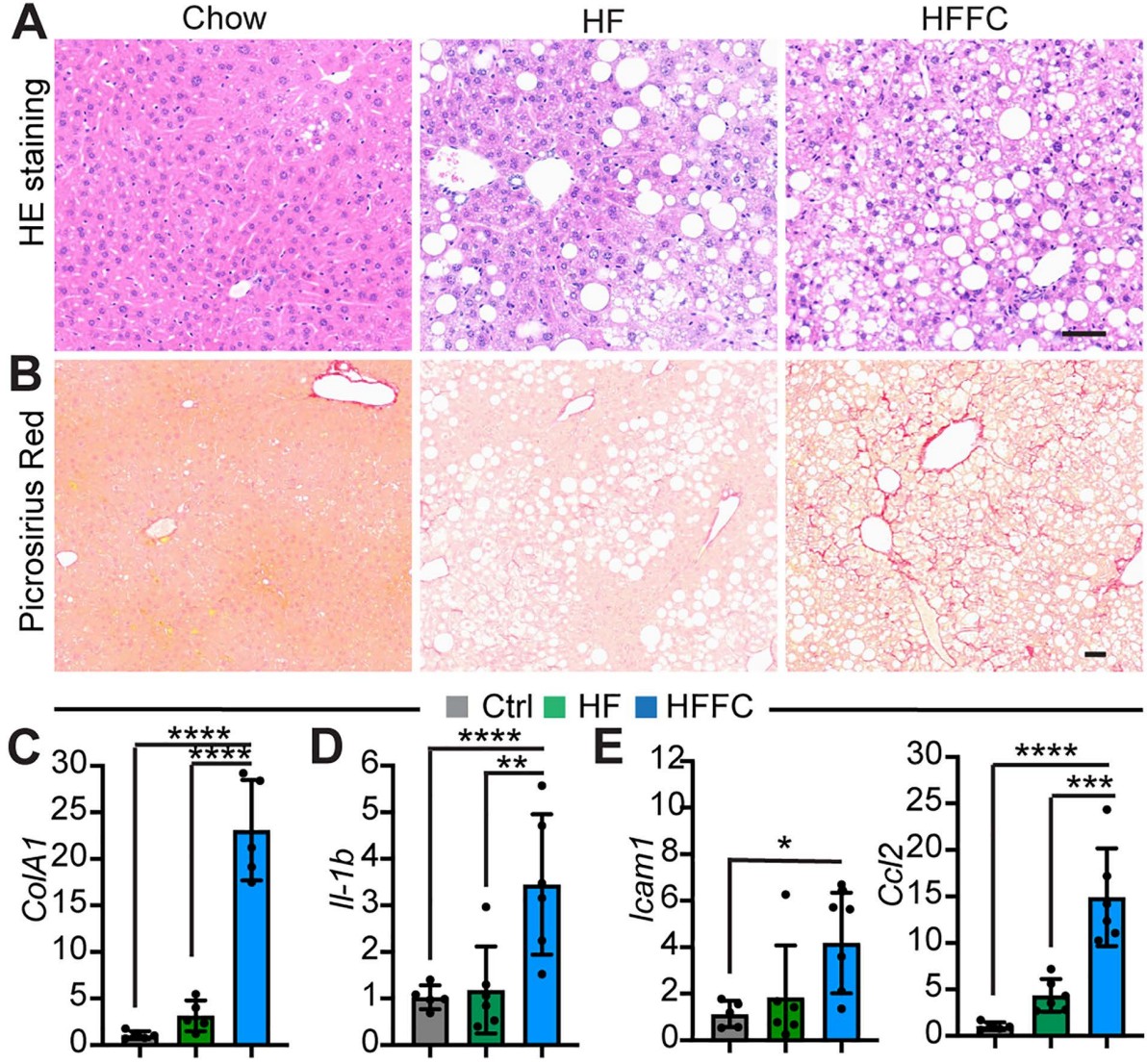

**Fig 4. Chronic HFFC diet treatment causes MASH. A.** HE staining of liver sections. **B.** Picrosirius Red staining of liver sections. A and B, scale: 50 µm. **C-E.** Quantitative PCR analysis of gene expression. One way ANOVA test, *, $p < 0.05$; **, $p < 0.01$; ***, $p < 0.001$; ****, $p < 0.0001$. N = 6.

(FS%) (Fig 5A), left ventricle wall thickness (Fig 5B), left ventricle chamber diameter (Fig 5C), and heart rate (Fig 5D). These data suggest that chronic HFFC diet treatment did not affect cardiac structure and systolic function. Second, we examined the cardiac diastolic function through mitral valve pulse-wave Doppler echocardiography. The ratio between the early diastolic filling (E-wave) and late diastolic filling (A-wave) velocity was a standard parameter for measuring cardiac diastolic function [22], which was not distinguishable between the Chow diet- and the HFFC diet-fed mice (Fig 5E and 5F).

In line with the previously published data [10], the HF diet treatment did not change cardiac diastolic function (S1 Fig). These echocardiography data together demonstrated that neither the HF diet nor the HFFC diet caused HFpEF.

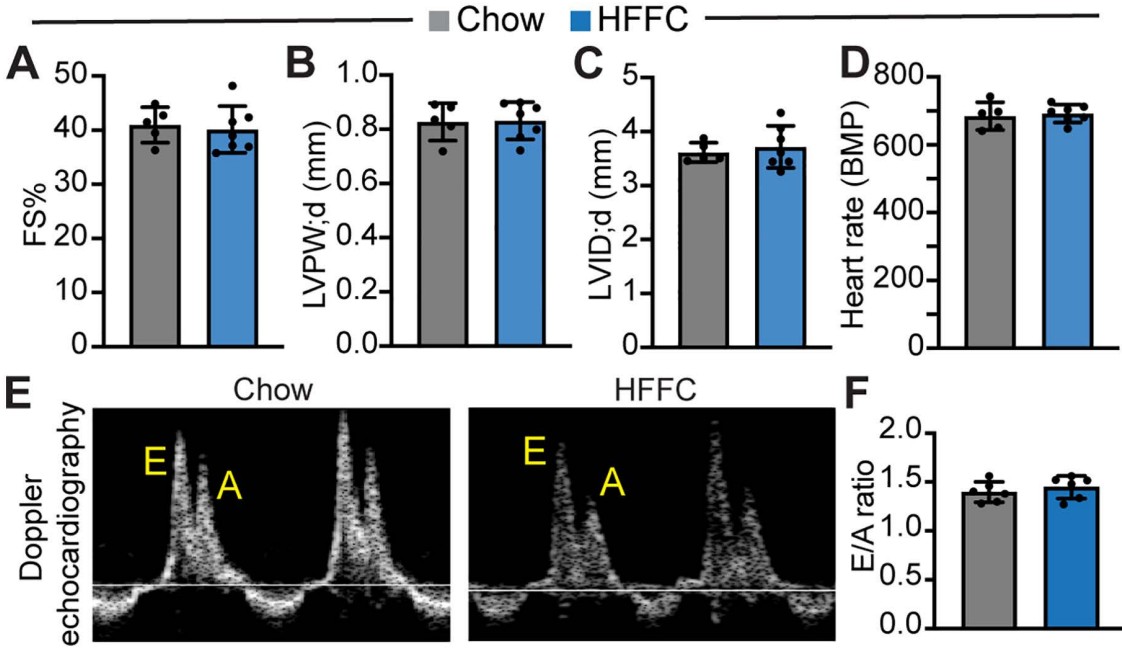

**Fig 5. Echocardiography measurements. A**. Fraction shortening. **B**. Left ventricle posterior wall thickness (LVPW), diastolic. **C**. Left ventricular internal diameter (LVID), diastolic. **D**. Heart rate. A, B, C and D, M-mode echocardiography measurements were performed with conscious mice. N = 6. **E**. Representative images of Doppler echocardiography measuring the peak velocities of the early diastolic filling (E-wave) and late diastolic filling (A-wave) across the mitral valve. **F**. Early to late diastolic transmitral flow velocity ratio (E/A). N = 6.

## The heart is tolerant to the HF diet and the HFFC diet stress

To further assess whether the HF diet and the HFFC diet stress affected the heart, we collected hearts and examined myocardial histology. Compared to the Chow diet, the HFFC diet treatment did not change the heart size (Fig 6A) and heart weight (Fig 6B). Because both the HF diet and the HFFC diet increased body weight, the heart to body weight ratio was decreased in these two diet-stressed groups (Fig 6C). The HF diet alone is known not to cause heart morphological changes [10], and the effects of the HFFC diet on the heart has not been addressed; therefore, we performed histological analysis with the hearts of the HFFC diet stressed mice. Apical four-chamber view of the hearts did not show atrial and ventricular morphological differences between the Chow and the HFFC hearts (Fig 6D). Additionally, the myocardium of neither the Chow nor the HFFC heart showed noticeable pathological changes (Fig 6E), suggesting that the HFFC diet is not harmful to the cardiomyocytes.

## Chronic HFFC diet treatment does not induce pathological cardiac hypertrophic remodeling

To further corroborate the histological observations, we prepared cardiac cross-sections and stained the myocardium with wheat germ agglutinin (WGA), a lectin that binds to basement membranes and fibrotic tissue [23]. The whole mount view of the cardiac cross-sections did not show WGA signal difference between the Chow, HF, and HFFC hearts (Figs 7A and S2A), and quantification of the septum thickness revealed no statistical difference between these three groups of hearts (Figs 7B and S2B).

Clinical studies have shown that cardiac papillary muscle abnormality is associated with hypertrophic cardiomyopathy and may precede the development of myocardial hypertrophy [24]. We then imaged and measured the papillary cardiomyocyte cross-sectional area to test whether the HF and the HFFC hearts were at an early stage of hypertrophy. The

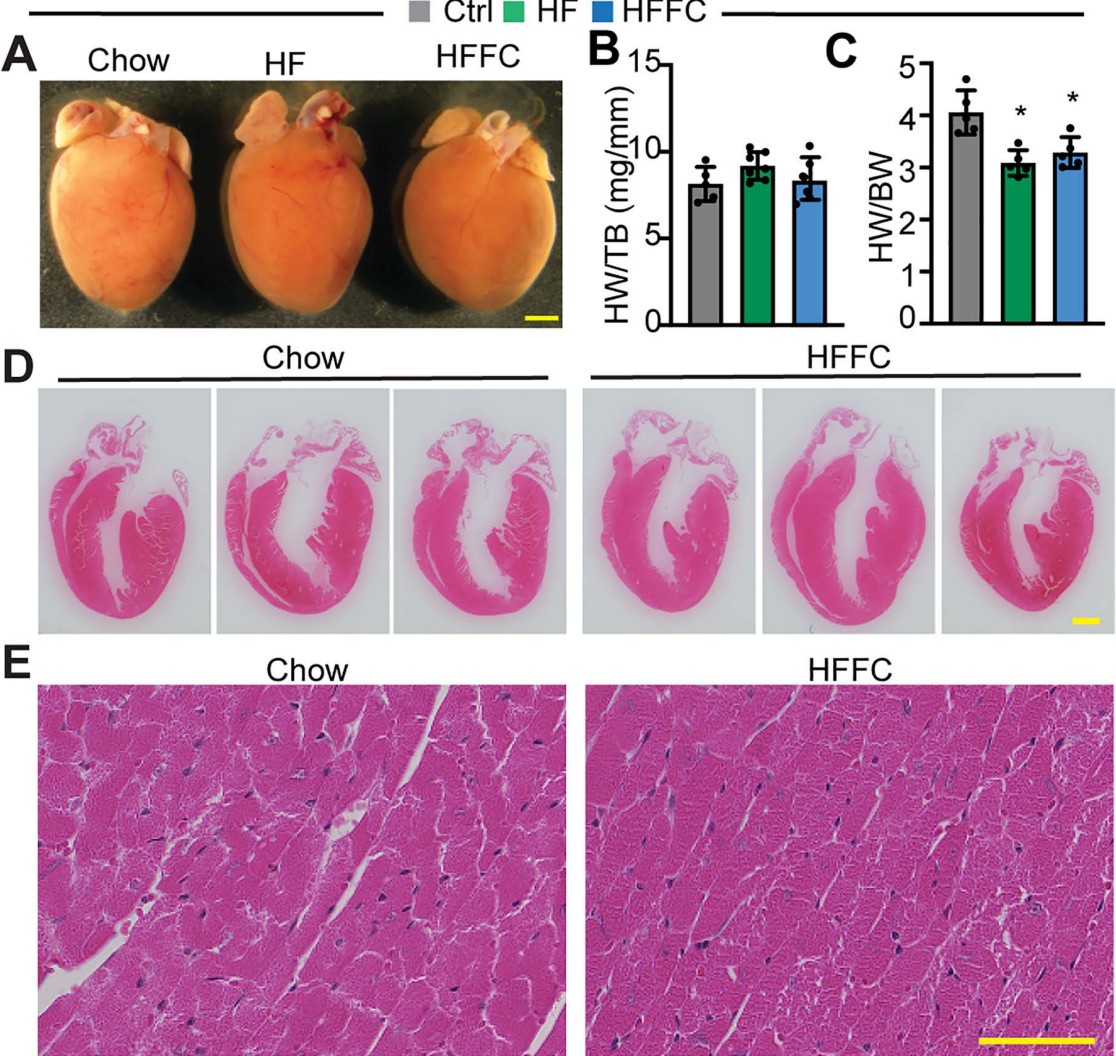

**Fig 6. Heart histology analysis. A**. Gross morphology of the heart. Scale, 1 mm. **B**. Heart weight to tibia bone length ratio. **C**. Heart to body weight ratio. B and C, n = 5. Student t-test, *, p < 0.05. **D**. Hematoxylin and eosin-stained cardiac longitudinal sections. Scale, 1 mm. **E**. Hematoxylin and eosin-stained myocardium. Scale, 50 μm.

results showed that both the HF and the HFFC papillary muscles had larger cardiomyocytes than their Chow counterparts (Figs 7C, 7D and S2C), suggesting cardiac hypertrophy. To further investigate whether this cardiac histological change was physiological or pathological, we examined the expression of four genes associated with pathological cardiac hypertrophy, including *Myosin heavy chain 6 (Myh6), Natriuretic Peptide A (Nppa)*, *Nppb,* and *Myh7* [25]. *Myh6* is decreased, whereas *Nppa*, *Nppb,* and *Myh7* are increased when the rodent heart is undergoing pathological hypertrophic remodeling [25]. We found that neither *Myh6* nor *Myh7* expression was altered by the HF diet or the HFFC diet; however, *Nppa* showed a decrease trend in the HFFC heart, and *Nppb* was significantly increased in the HF heart (Figs 7E and S2D). On the protein level, MYH6 expression was similar between the Chow diet- and the HFFC diet-fed mice (Fig 7F and 7G).

Altogether, these data suggest that either HF diet or HFFC diet treatment induces an early stage of hypertrophy, which is unlikely to be pathological.

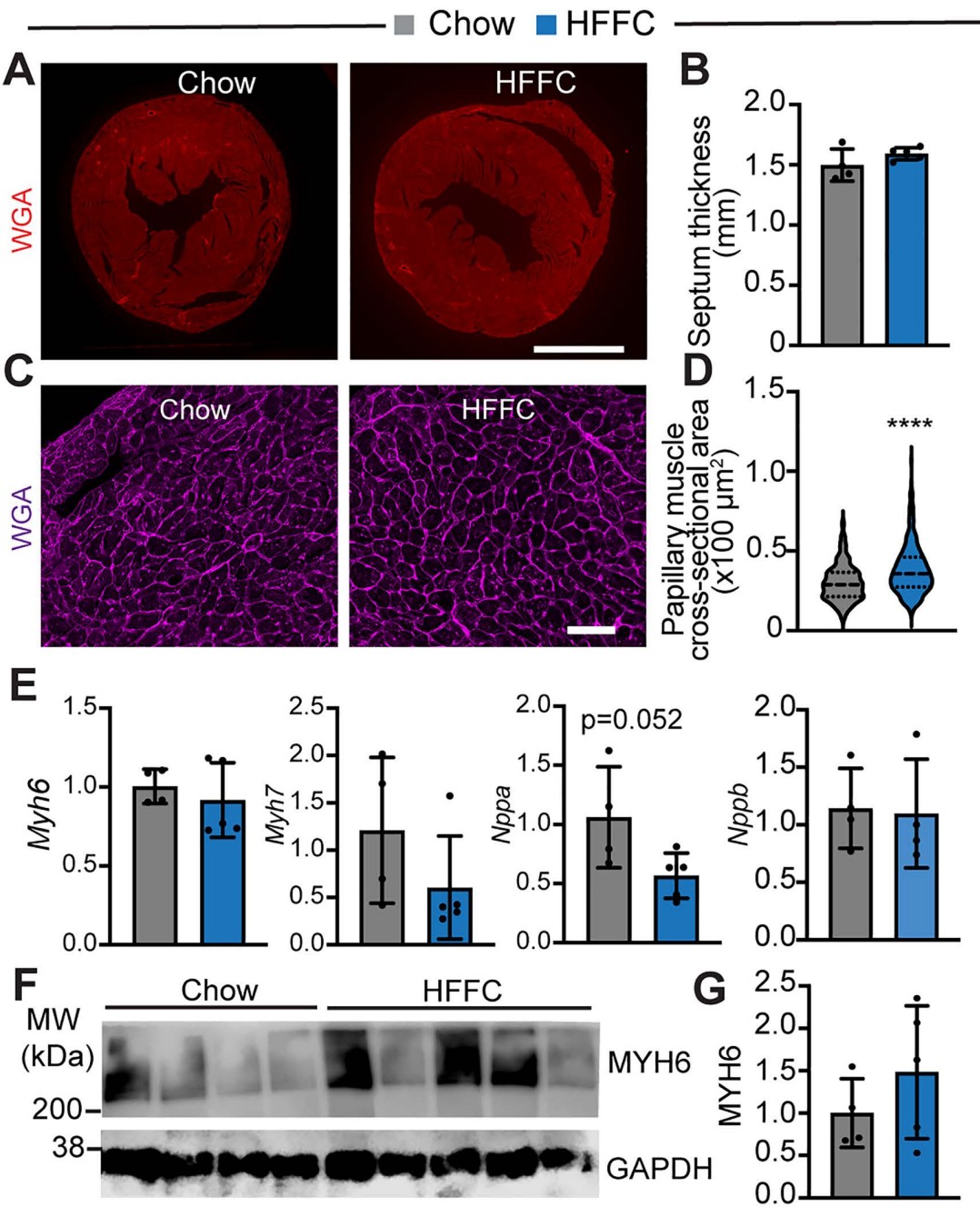

**Fig 7. Cardiac hypertrophy analysis. A**. Wheat germ agglutinin (WGA)-stained cardiac cross sections. Scale, 2 mm. **B**. Cardiac septum thickness measurement. Chow, n = 4; HFFC, n = 5. **C**. WGA-stained papillary muscle cross sections. Scale, 50 μm. **D**. Papillary muscle cross-sectional area measurement. Chow diet group, 200 cells from 4 hearts were measured. HFFC diet group, 250 cells from 5 hearts were measured. Mann-Whitney test, ****, p < 0.0001. **E**. Quantitative PCR analysis of gene expression. Chow, n = 4; HFFC, n = 5. **F**. MYH6 Western blot. GAPDH was used as a loading control. **G**. Densitometry quantification MYH6. Chow, n = 4; HFFC, n = 5.

## Discussion

Both the HF diet and the HFFC diet have been used to study MAFLD in mice, with the HFFC diet (also known as the Western diet or AMLN diet) inducing a liver condition that aligns closely with human MASH [26]; however, it is not clear whether these two diets cause obesity to the same extent. Our current work demonstrates that the HF diet is more efficient than the HFFC diet for inducing obesity/adipose tissue expansion, but less potent than the HFFC diet for inducing liver steatosis, inflammation, and fibrosis. Our observations further support the adipose tissue expandability hypothesis, which states that failed adipose tissue expansion leads to ectopic lipid accumulation in non-adipocyte cells, followed by tissue inflammation [27].

Besides adipose tissue, the liver is another critical organ with significant lipid storage capacity and communicates with adipose tissue to regulate lipid metabolism [28]. After intestinal absorption, triglycerides from the food are transported to adipose tissue for storage and to the liver for further processing. In the liver, lipids can be packaged into very low-density lipoprotein (VLDL) for secretion or being stored in the hepatocytes [29]. In our current study, we used sex- and age-matched C57BL/6N mice to test the metabolic effects of two formulated diets. During the same period of treatment, the HFFC diet induced less adipose tissue expansion and more severe liver steatosis than the HF diet, thus suggesting that lipids from the HFFC diet could not be efficiently stored in the adipose tissue. Consequently, lipids from the HFFC diet were deposited mainly in the liver. These metabolic effects comparisons between the HF diet and HFFC diet raise two fundamental questions: 1) Is the cholesterol or the fructose of the HFFC diet responsible for damaging the liver? 2) Do these two nutrients suppress adipose tissue expansion? Addressing these two questions will provide new insights for understanding how the liver and adipose tissue orchestrate metabolic homeostasis.

Our current data demonstrate that the HFFC diet-induced MASH is not associated with HFpEF in middle-aged mice. Pioneer work from the Joseph A. Hill group has shown that HF diet treatment efficiently causes obesity and fatty liver disease without triggering hypertension and HFpEF in mice, and that the addition of L-NAME stress to HF diet-fed mice successfully induces HFpEF [10], highlighting the causal role of hypertension in the etiology of HFpEF. Although obesity and fatty liver disease do not increase blood pressure in mice [10], these metabolic conditions are highly associated with increased risk of hypertension in human patients [30,31]. This interspecies difference may be because rodents have a higher rate of cholesterol and bile acid metabolism and a lower serum LDL level than humans [32]. Different from mice, hamsters have a lipoprotein metabolism profile similar to humans [33], and hamsters tend to develop hypertension when stressed with a HF diet [34]. Consistently, published data have shown that hamsters developed both MASH and HFpEF when stressed by a high-fat, high-fructose diet [12], further suggesting that MASH and hypertension are both required to cause HFpEF.

The HFFC diet does not harm the cardiomyocytes. The HFFC diet has been widely used to study MASH in rodents; however, the cardiac function and histopathology of these MASH mice have not been characterized. In this study, we did not find significant cardiac pathological changes in the HFFC-treated mice, but we did notice a mild increase in papillary muscle cell size. The observed increase in papillary muscle cell size is a physiological rather than a pathological hypertrophic response, as evidenced by the largely unchanged expression of pathological hypertrophic marker genes. Our results are consistent with a recently published study, which shows that a choline-deficient (CDAA) diet induces MASH without affecting the heart in middle-aged C57BL/6J (10 months old) mice [35]. Although the C57BL/6N strain is different from the C57BL/6J strain that carries a deficient nicotinamide nucleotide transhydrogenase (NMT) gene, the susceptibility to HF diet-induced metabolic dysfunction is similar between 6J and 6N sub-strains [36]. Our current study, together with the published data, suggests that MASH does not cause cardiac pathological changes in middle-aged C57BL/6N or C57BL/6J mice.

A high-fat (35.5%) and high-sucrose (36.3%) diet (HFHS) is another type of western diet used to study metabolic syndrome, and chronic treatment of mice with the HFHS diet causes pathological cardiac hypertrophy and diastolic dysfunction [37], but rarely leads to MASH [38]. The HFFC diet used in the current study contains fructose rather than sucrose

as the carbohydrate source, suggesting that sucrose, rather than fructose or fat, is the nutrient responsible for cardiac pathological hypertrophic remodeling in obese patients. This hypothesis is in line with a recent study showing that increasing glucose consumption enhances cardiac hypertrophic growth [39], and is also supported by the observation that HF diet prevents hypertension-induced cardiac hypertrophy in rats [40].

## Conclusions

By comparing the pathophysiological effects of two formulated diets in mice, we found that the HF diet is more potent than the HFFC diet in inducing obesity. *In contrast, the HFFC* diet is more toxic to the liver than the HF diet. Chronic HFFC diet treatment of mice resulted in severe steatohepatitis, which is not associated with cardiac diseases.

## Supporting information

**S1 Table. Primers used in this study.**
(PDF)

**S1 Fig. Related to** Fig 5**.** Early to late diastolic transmitral flow velocity ratio (E/A). N = 6.
(TIF)

**S2 Fig. Related to** Fig 7**. A**. WGA-stained cardiac cross sections. Scale, 2 mm. **B**. Cardiac septum thickness measurement. **C**. Papillary muscle cross-sectional area measurement. Chow diet group, 200 cells from 4 hearts were measured. For each group of the high-fat diet (HF) and HFFC diet-treated mice, 250 cells from 5 hearts were measured. Kruskal-Wallis test, ****, p < 0.0001. **D**. Quantitative PCR analysis of heart gene expression. Student t-test, *, p < 0.05. N = 5.
(TIF)

**S1 Raw Images. Original western blot images for** Fig 7F**.**
(TIF)

## Author contributions

**Conceptualization:** Zhiqiang Lin.

**Data curation:** Zhiqiang Lin.

**Formal analysis:** Zhiqiang Lin, Chase W. Kessinger, Lin Chen, Felicia Huang, Jierui Lin.

**Funding acquisition:** Zhiqiang Lin, Chase W. Kessinger.

**Investigation:** Zhiqiang Lin, Chase W. Kessinger, Lin Chen, Felicia Huang, Jierui Lin, Amanda Davenport.

**Methodology:** Chase W. Kessinger.

**Writing – original draft:** Zhiqiang Lin.

**Writing – review & editing:** Chase W. Kessinger, Felicia Huang.

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
