## [Decision Letter · Decision Letter 0]

11 Sep 2025

Dear Dr. Lin,

Thank you for submitting your manuscript to PLOS ONE. After careful consideration, we feel that it has merit but does not fully meet PLOS ONE’s publication criteria as it currently stands. Therefore, we invite you to submit a revised version of the manuscript that addresses the points raised during the review process.

We look forward to receiving your revised manuscript.

Kind regards,

Rami Salim Najjar, Ph.D.

Academic Editor

PLOS ONE

Journal Requirements:

2. To comply with PLOS One submissions requirements, in your Methods section, please provide additional information regarding the experiments involving animals and ensure you have included details on (1) methods of sacrifice, (2) methods of anesthesia and/or analgesia, and (3) efforts to alleviate suffering.

4. Thank you for stating the following financial disclosure: [Z. L. was supported by NHLBI (HL146810), Taconic research grant and AHA Transformational Project Award (25TPA1478779).]. 

5. Please expand the acronym “NHLBI” and “AHA”(as indicated in your financial disclosure) so that it states the name of your funders in full.

6. Thank you for stating the following in your Competing Interests section: [None].

8. PLOS ONE now requires that authors provide the original uncropped and unadjusted images underlying all blot or gel results reported in a submission’s figures or Supporting Information files. This policy and the journal’s other requirements for blot/gel reporting and figure preparation are described in detail at https://journals.plos.org/plosone/s/figures#loc-blot-and-gel-reporting-requirements and https://journals.plos.org/plosone/s/figures#loc-preparing-figures-from-image-files. When you submit your revised manuscript, please ensure that your figures adhere fully to these guidelines and provide the original underlying images for all blot or gel data reported in your submission. See the following link for instructions on providing the original image data: https://journals.plos.org/plosone/s/figures#loc-original-images-for-blots-and-gels.  

Reviewers' comments:

Reviewer's Responses to Questions

**Comments to the Author**

1. Is the manuscript technically sound, and do the data support the conclusions?

Reviewer #1: Partly

Reviewer #2: Partly

2. Has the statistical analysis been performed appropriately and rigorously?

Reviewer #1: I Don't Know

Reviewer #2: Yes

3. Have the authors made all data underlying the findings in their manuscript fully available?

Reviewer #1: Yes

Reviewer #2: Yes

4. Is the manuscript presented in an intelligible fashion and written in standard English?

Reviewer #1: Yes

Reviewer #2: Yes

Reviewer #1: Manuscript number: PONE-D-25-42066

Title: Diet-induced steatohepatitis does not cause heart failure with preserved ejection function in middle-aged mice.

Summary: This manuscript describes possible path to development of HFpEF in MASH patients by treating middle-aged mice with HF or HFFC diet. Mice were fed two different diets to induce obesity, metabolic dysfunction, and steatohepatitis. Mice were tested using an array of imaging, biochemical, and histological tests to test their hypothesis. Although both diets increased body weights, fasting glucose levels, and liver and hepatocyte size, HFFC diet treated mice demonstrated more severe steatohepatitis than HF diet alone. Mice on HFFC diet, however, did not develop impaired cardiac function or histopathological changes indicative of HFpEF.

This manuscript has potential, and the amount of work that went into it is evident. However, I have some comments that I hope will improve it and make it more scientifically well-rounded.

General comments:

1. Authors need to thoroughly revise manuscript for grammar, formatting, and accuracy. There were several instances where I found misplaced or extra spaces, use of incorrect wording and sentence structure, and inaccurate statements.

2. Authors define HFpEF as heart failure with preserved ejection function, but the most accurate and standard definition of HFpEF is heart failure with preserved ejection fraction.

Specific comments:

1. Title would benefit from adding sex and type of mouse used, in addition to changing definition of HFpEF as stated above.

2. Heart failure is not only defined by the heart being unable to “adequately discharge its contents” (lines 32-33). Please expand/provide complete definition of heart failure.

3. Authors state that “No evidence-based therapeutics have been developed to treat HFpEF” (lines 35-36). This statement is not accurate. They cite Schiattarella et al. (2019), to sustain claim, but this manuscript only expands on the failure of NO-inducing approaches as HFpEF therapeutics. While HFpEF has historically been challenging to treat, there are evidence-based therapeutics available and in progress. This needs to be discussed.

4. Although it is true that animal models can provide valuable mechanistic insights into the relationship between MASH and HFpEF, they are not necessarily critical for defining this relationship. Rewrite sentence in lines 41-42 to better reflect the utility of animal models in HFpEF research.

5. The experiments were conducted in males only. There is no background or justification for this provided in the introduction or discussion. Since female sex is a risk factor for HFpEF, justification for using only males should be provided.

6. There is no discussion on why the hamster MASH model developed HFpEF, and why the authors decided to replicate in mice. Leap in logic for readers (lines 49-52).

7. Line 50. Misspelled HFpEF as HEpEF.

8. Authors state “…thus raising the question of whether murine MASH model could be used for studying HFpEF.” (Lines 51-52). Murine models have been used to study HFpEF. Please clarify intent here.

9. The word “specify” is not used correctly in line 53. Authors may want to replace with “clarify”, “elucidate”, “identify”, “determine” or another alternative.

10. Reword sentence in line 57-59. It is not clear why authors studied the metabolic pathophysiological effects of two diets. State purpose of study directly. Where the diets used to determine whether HFpEF is induced by steatohepatitis alone?

11. Please provide sex, age, and total n per group in the animals section of the methodology.

12. Statement in line 146 may be an overreach, hyperglycemia alone does not confirm metabolic dysfunction. Suggest attenuating language here.

13. Although authors state that because HFFC diet more effectively led to hepatic steatosis in figure 3, they only continued with HFFC group (lines 182-185), it would have been beneficial to continue experiments with HF group as well. There may have been cardiac associated effects observed with HF that were not developed in HFFC treated mice. Suggest adding more data from HF group or expanding on the discussion.

14. Figure 3E is missing chow fed control data.

15. Did the authors consider inducing hypertension in their MASH model given their understanding of other mouse models of HFpEF? Please discuss. What differences between hamsters and mice could lead to differences in development of HFpEF after HFFC diet? What about differences between mice, hamsters, and human? Expand on translatability of these models.

16. Was blood pressure tested in these groups? Authors state in lines 48-49 that “the introduction of chronic hypertension is essential for the metabolic dysfunctional mice to develop HFpEF” but do not discuss further. They need to provide data and/or discuss blood pressure effects of both HF diet and HFFC diet in mice to complete assessment of their model.

17. Please explain in more detail why the findings on lines 248-251 were interesting/not expected. Authors suggest in lines 253-257 that this effect has been observed in the literature (Clapper et al, 2013, and Matsumoto et al, 2013), and it is well known that HFFC is more detrimental to liver than HF alone.

18. Add more citations to discussion in lines 251-260.

19. Manuscript would benefit from a statement disclosing criteria/reasons for excluding animals from each study. Differing n in figure legends is confusing.

Reviewer #2: This study demonstrates that a HFFC diet induces MASH with mild fibrosis but not cardiac disease.

1. Cardiovascular disease correlates with fibrosis stage. This model generates stage 1-2 fibrosis, so cardiac disease would not be expected.

2. What do the mice die from? Liver disease or heart disease. How does the age of death compare to control mice?

How does the different strains of mice correlate to generation of liver disease and cardiac disease? Why was this strain choses and does this strain develop cardiac disease in other models?

3. Are there any biomarkers of heart disease that might proceed frank disease? What is the BNP level, what about the transcriptomics and pathway analysis.

**Do you want your identity to be public for this peer review?** For information about this choice, including consent withdrawal, please see our Privacy Policy

Reviewer #1: No

Reviewer #2: No

---

## [Author Response · Author response to Decision Letter 1]

13 Nov 2025

Response to Reviewers’ Comments for PONE-D-25-42066 entitled “Diet-induced steatohepatitis does not cause heart failure with preserved ejection function in middle-aged mice”

We would like to express our gratitude to the editors and reviewers for their interest and valuable suggestions regarding our research article. We hope that our revised manuscript and responses meet the expectations and requirements of all reviewers and editors, as well as the journal's standards.

Reviewer #1 Comments and Responses:

Summary: This manuscript describes possible path to development of HFpEF in MASH patients by treating middle-aged mice with HF or HFFC diet. Mice were fed two different diets to induce obesity, metabolic dysfunction, and steatohepatitis. Mice were tested using an array of imaging, biochemical, and histological tests to test their hypothesis. Although both diets increased body weights, fasting glucose levels, and liver and hepatocyte size, HFFC diet treated mice demonstrated more severe steatohepatitis than HF diet alone. Mice on HFFC diet, however, did not develop impaired cardiac function or histopathological changes indicative of HFpEF. This manuscript has potential, and the amount of work that went into it is evident. However, I have some comments that I hope will improve it and make it more scientifically well-rounded.

We are very grateful for your acknowledgements and comments aimed to improve our manuscript.

General Comments:

1. Authors need to thoroughly revise manuscript for grammar, formatting, and accuracy. There were several instances where I found misplaced or extra spaces, use of incorrect wording and sentence structure, and inaccurate statements.

Thank you for your comment. We have reviewed and revised the manuscript to correct these insufficiencies.

2. Authors define HFpEF as heart failure with preserved ejection function, but the most accurate and standard definition of HFpEF is heart failure with preserved ejection fraction.

Thank you for your comment. We have made the changes and apologize for our oversight.

Specific comments:

1. Title would benefit from adding sex and type of mouse used, in addition to changing the definition of HFpEF as stated above.

Thank you for your comment. We agree with the reviewer that the inclusion of sex and mouse line will be helpful in future referencing of this work in the field. We have now changed the title to “Diet-induced steatohepatitis does not cause heart failure with preserved ejection fraction in male middle-aged C57BL/6N mice.”

2. Heart failure is not only defined by the heart being unable to “adequately discharge its contents” (lines 32-33). Please expand/provide a complete definition of heart failure.

Thank you for your comment. We have expanded the text in the manuscript, and it is provided below for your convenience.

“Heart failure is a condition characterized by the inability of the ventricles to fill or eject blood, resulting in inadequate cardiac output. It can arise from changes in the structure or function of the myocardium, valves, and vessels of the heart, and is clinically classified as two distinct forms: heart failure with reduced ejection fraction (HFrEF) and heart failure with preserved ejection fraction (HFpEF).”

3. Authors state that “No evidence-based therapeutics have been developed to treat HFpEF” (lines 35-36). This statement is not accurate. They cite Schiattarella et al. (2019), to sustain claim, but this manuscript only expands on the failure of NO-inducing approaches as HFpEF therapeutics. While HFpEF has historically been challenging to treat, there are evidence-based therapeutics available and in progress. This needs to be discussed.

We thank the reviewer for their suggestion and have added text to discuss the recent progress. We have provided the addition below for your convenience.

“Although many medicines for managing HFrEF are available, only a few therapeutics treating HFpEF have been developed in recent years (see review [7]).”

4. Although it is true that animal models can provide valuable mechanistic insights into the relationship between MASH and HFpEF, they are not necessarily critical for defining this relationship. Rewrite sentence in lines 41-42 to better reflect the utility of animal models in HFpEF research.

We thank the reviewer for their comment and have adjusted our text to highlight the utility of the animal models in HFpEF research. We have provided the text below for your convenience.

“Animal models are critical for defining the molecular mechanisms that link MASH and HFpEF.”

5. The experiments were conducted in males only. There is no background or justification for this provided in the introduction or discussion. Since female sex is a risk factor for HFpEF, justification for using only males should be provided.

Thank you for your comment. We provided several sentences to justify the use of male mice.

“Besides genetic influence, sex is another risk factor for MASH. For example, middle-aged male patients more likely to develop MASH than age-matched female patients [13], and this sex dimorphism is recapitulated in diet-induced murine MASH models [14,15]. For this scientific reason, we chose to use male mice in the current study.”

6. There is no discussion on why the hamster MASH model developed HFpEF, and why the authors decided to replicate in mice. Leap in logic for readers (lines 49-52).

We thank the reviewer for their comment. In the discussion section, we put a few sentences to explain the differences between hamster and mouse MASH models.

“Although obesity and fatty liver disease do not increase blood pressure in mice [10], these metabolic conditions are highly associated with increased risk of hypertension in human patients [31,32]. This interspecies difference may be because rodents have a higher rate of cholesterol and bile acid metabolism and a lower serum LDL level than humans. Different from mice, hamsters have a lipoprotein metabolism profile similar to humans [33], and hamsters tend to develop hypertension when stressed with a HF diet [34]. Consistently, published data have shown that hamsters developed both MASH and HFpEF when stressed by a high-fat, high-fructose diet [12], further suggesting that MASH and hypertension are both required to cause HFpEF.”

In the introduction section, two sentence were added to make the logic flows smoothly:

“The relationship between HFpEF and MASH has been studied in a high fat/high cholesterol diet-induced hamster MASH model, which shows a clear correlation between steatohepatitis and HFpEF [12]; however, the lack of genetically modified hamster strains limits the use of this hamster MASH model for studying the molecular mechanisms linking MASH with HFpEF. In contrast, a large collection of gene modified mouse lines are publicly available and are an invaluable asset for deciphering the genetic factors controlling disease progression, thus it is a high impact question to determine whether murine MASH model could be used for studying HFpEF.”

7. Line 50. Misspelled HFpEF as HEpEF.

Thank you for your comment. We have corrected the misspelling of HEpEF to HFpEF.

8. Authors state “…thus raising the question of whether murine MASH model could be used for studying HFpEF.” (Lines 51-52). Murine models have been used to study HFpEF. Please clarify intent here.

Thank you for your comment. Ref question 6.

9. The word “specify” is not used correctly in line 53. Authors may want to replace with “clarify”, “elucidate”, “identify”, “determine” or another alternative.

Thank you for your comment. We have revised the text and removed the sentence containing this word.

10. Reword sentence in line 57-59. It is not clear why authors studied the metabolic pathophysiological effects of two diets. State purpose of the study directly. Where the diets used to determine whether HFpEF is induced by steatohepatitis alone?

We revised the text to clarify the purpose of the current study: “To establish a vigorous diet-induced MASH model for studying HFpEF, we compared the pathological effects of two formulated diets for triggering MASH, including a high fat (HF) diet, and a high fat, high fructose, high cholesterol diet (HFFC, also known as AMLN diet [16]). We found that both the HF diet and the HFFC diet treatment induced liver steatosis, but only the HFFC diet caused MASH. By characterizing the cardiac phenotypes of these HF diet and HFFC diet treated mice, we concluded that neither liver steatosis nor MASH caused HFpEF in middle-aged male C57/BL6N mice.”

11. Please provide sex, age, and total n per group in the animals section of the methodology.

This information has been provided in the Diet stress and tissue collection

“The diet stress was started with 8-weeks-old male C57BL/6N mice, and they were sacrificed for tissue collection 36 weeks after diet stress. For each cohort, 8 animals were included in the beginning of the study. For tissue collection, mice were first sacrificed with CO2 overdose, and cardiac perfusion was immediately followed to remove blood from the tissue. For the following studies, 4-6 successfully collected liver or heart samples were included in the histology and molecular analysis”.

12. Statement in line 146 may be an overreach, hyperglycemia alone does not confirm metabolic dysfunction. Suggest attenuating language here.

The sentence has been revised:

“Both the HF diet and the HHFC diet induced hyperglycemia (Fig. 1F), suggestive of metabolic dysfunction in these mice.”

13. Although authors state that because HFFC diet more effectively led to hepatic steatosis in figure 3, they only continued with HFFC group (lines 182-185), it would have been beneficial to continue experiments with HF group as well. There may have been cardiac associated effects observed with HF that were not developed in HFFC treated mice. Suggest adding more data from HF group or expanding on the discussion.

We did liver and cardiac analysis with the HF group. The HF group data were provided in the revised manuscript. The new data can be found in the revised Figure 4, Figure 6, supplementary Figure 1 and supplementary Figure 2.

14. Figure 3E is missing chow fed control data.

When we measured the hepatocyte lipid droplets, the chow diet control hepatocytes contained no meaningful large lipid droplets, so the chow fed control data were not included.

15. Did the authors consider inducing hypertension in their MASH model given their understanding of other mouse models of HFpEF? Please discuss. What differences between hamsters and mice could lead to differences in development of HFpEF after HFFC diet? What about differences between mice, hamsters, and human? Expand on translatability of these models.

Response: In this work we tried to assess whether MASH alone causes HFpEF, so we did not consider inducing hypertension in these mice.

In the discussion section, we added in a paragraph to address these issues:

“Our current data demonstrate that HFFC diet-induced MASH is not associated with HFpEF in middle-aged mice. Pioneer work from the Joseph A. Hill group has shown that HF diet treatment efficiently causes obesity and fatty liver disease without triggering hypertension and HFpEF in mice, and that the addition of L-NAME stress to HF diet-fed mice successfully induces HFpEF [30], highlighting the causal role of hypertension in the etiology of HFpEF. Although obesity and fatty liver disease do not increase blood pressure in mice [10], these metabolic conditions are highly associated with increased risk of hypertension in human patients [31,32]. This interspecies difference may be because rodents have a higher rate of cholesterol and bile acid metabolism and a lower serum LDL level than humans. Different from mice, hamsters have a lipoprotein metabolism profile similar to humans [33], and hamsters tend to develop hypertension when stressed with a HF diet [34]. Consistently, published data have shown that hamsters developed both MASH and HFpEF when stressed by a high-fat, high-fructose diet [12], further suggesting that MASH and hypertension are both required to cause HFpEF.”

16. Was blood pressure tested in these groups? Authors state in lines 48-49 that “the introduction of chronic hypertension is essential for the metabolic dysfunctional mice to develop HFpEF” but do not discuss further. They need to provide data and/or discuss blood pressure effects of both HF diet and HFFC diet in mice to complete assessment of their model.

Response: we did not test the blood pressure because we were short of instruments for measuring blood pressure during our study.

Ref question 15 for the detail explanation.

17. Please explain in more detail why the findings on lines 248-251 were interesting/not expected. Authors suggest in lines 253-257 that this effect has been observed in the literature (Clapper et al, 2013, and Matsumoto et al, 2013), and it is well known that HFFC is more detrimental to liver than HF alone.

Response: We revised the discussion to explain significance of the new findings.

“Both HF and HFFC diets have been used to study MAFLD in mice, with the HFFC diet (also known as the Western diet or AMLN diet) inducing a liver condition that aligns closely with human MASH [26]; however, it is not clear whether these two diets cause obesity to the same extent. Our current work demonstrates that the HF diet is more efficient than the HFFC diet for inducing obesity/adipose tissue expansion, but less potent than the HFFC diet for inducing liver steatosis, inflammation, and fibrosis. Our observations further support the adipose tissue expandability hypothesis, which states that failed adipose tissue expansion leads to ectopic lipid accumulation in non-adipocyte cells, followed by tissue inflammation [27].”

18. Add more citations to discussion in lines 251-260.

We revised this discussion part to include more information.

“Besides adipose tissue, the liver is another critical organ with significant lipid storage capacity and communicates with adipose tissue to regulate lipid metabolism [28]. After intestinal absorption, triglycerides from the food are transported to adipose tissue for storage and to the liver for further processing. In the liver, lipids can be consumed or packaged into very low-density lipoprotein (VLDL) for secretion, or stored in the hepatocytes [29]. In our current study, we used sex- and age-matched C57BL/6N mice to test the metabolic effects of two formulated diets. During the same period of treatment, the HFFC diet induced less adipose tissue expansion and more severe liver steatosis than the HF diet, thus suggesting that lipids from the HFFC diet could not be efficiently stored in the adipose tissue. Consequently, lipids from the HFFC diet were deposited mainly in the liver. These metabolic effects comparisons between the HF diet and HFFC diet raise two fundamental questions: 1) Is the cholesterol or the fructose of the HFFC diet responsible for damaging the liver? 2) Do these two nutrients suppress adipose tissue expansion? Addressing these two questions will provide new insights for understanding how the liver and adipose tissue orchestrate metabolic homeostasis.”

19. Manuscript would benefit from a statement disclosing criteria/reasons for excluding animals from each study. Differing n in figure legends is confusing.

Please refer to the response to question 11.

Reviewer #2 Comments and Responses:

Reviewer #2: This study demonstrates that a HFFC diet induces MASH with mild fibrosis but not cardiac disease.

We thank you for your constructive comments, which will aid in improving our manuscript.

1. Cardiovascular disease correlates with fibrosis stage. This model generates stage 1 to stage 2 fibrosis, so cardiac disease would not be expected.

Response: We agree with the reviewer’s comments. We only observed mild hypertrophic responses with these two high fat diet models, and this hypertrophic response is likely physiologic rather than pathologic.

2. What do the mice die from? Liver disease or heart disease. How does the age of death compare to control mice?

How does the different strains of mice correlate to generation of liver disease and cardiac disease? Why was this strain chosen and does this strain develop cardiac disease in other models?

Response

---

## [Decision Letter · Decision Letter 1]

10 Dec 2025

Diet-induced steatohepatitis does not cause heart failure with preserved ejection fraction in male middle-aged C57BL/6N mice

PONE-D-25-42066R1

Dear Dr. Lin,

We’re pleased to inform you that your manuscript has been judged scientifically suitable for publication and will be formally accepted for publication once it meets all outstanding technical requirements.

Kind regards,

Rami Salim Najjar, Ph.D.

Academic Editor

PLOS One

Additional Editor Comments (optional):

Reviewers' comments:

Reviewer's Responses to Questions

**Comments to the Author**

Reviewer #1: All comments have been addressed

Reviewer #2: All comments have been addressed

2. Is the manuscript technically sound, and do the data support the conclusions?

Reviewer #1: Yes

Reviewer #2: Yes

3. Has the statistical analysis been performed appropriately and rigorously?

Reviewer #1: Yes

Reviewer #2: Yes

4. Have the authors made all data underlying the findings in their manuscript fully available?

Reviewer #1: Yes

Reviewer #2: Yes

5. Is the manuscript presented in an intelligible fashion and written in standard English?

Reviewer #1: Yes

Reviewer #2: Yes

Reviewer #1: (No Response)

Reviewer #2: The authors have adequately addressed the comments by myself and by the other reviewer. The limitations of this study are acknowledged.

**Do you want your identity to be public for this peer review?** For information about this choice, including consent withdrawal, please see our Privacy Policy

Reviewer #1: No

Reviewer #2: No

---

## [Editor Report · Acceptance letter]

PONE-D-25-42066R1

PLOS One

Dear Dr. Lin,

I'm pleased to inform you that your manuscript has been deemed suitable for publication in PLOS One. Congratulations! Your manuscript is now being handed over to our production team.

Kind regards,

on behalf of

Dr. Rami Salim Najjar

Academic Editor

PLOS One